



# Bottom mixed layer derivation and spatial variability over the central and eastern abyssal Pacific Ocean

Jessica Kolbusz[1,2], Devin Harrison[3], Nicole Jones[2,6], Joanne O'Callaghan[4,5], Taimoor Sohail[7], Todd Bond[1,2], Heather Stewart[3], and Alan Jamieson[1,2]

[1]School of Biological Sciences, Crawley, Australia
[2]Oceans Institute, University of Western Australia, Crawley, Australia
[3]Kelpie Geoscience Ltd., Edinburgh, United Kingdom
[4]Department of Physics, University of Auckland, Auckland, New Zealand
[5]Oceanly Science Limited, Wellington, New Zealand
[6]School of Earth and Environment, University of Western Australia, Crawley, Australia
[7]School of Geography, Earth and Atmospheric Science, University of Melbourne, Melbourne, Australia

**Correspondence:** Jessica Kolbusz (jess.kolbusz@uwa.edu.au)

**Abstract.** The bottom mixed layer (BML) of the abyssal ocean regulates heat exchange between the deep interior and seafloor, driving water-mass transformation and influencing global circulation. Spatial variability of the BML was examined in the under-sampled abyssal Pacific Ocean using surface-to-seafloor temperature and pressure observations over 4 months in 2023-24. Given the typical decadal repeat rate of global hydrographic sections, subdecadal variability in the abyssal ocean has remained poorly resolved. Our observations contribute towards filling this gap for the central and eastern abyssal Pacific Ocean. Four methods were used to determine the BML thickness, with the threshold method providing the most reliable estimates. The mean BML thickness was ($226 \pm 172$ m) with added repeat hydrographic sections providing context and additional data points. At each BML data point we determined the slope, the terrain roughness and the extracted predicted internal tide energy dissipation (over five different low-mode processes and high-mode local processes) at 50 km scales from publicly available datasets. These factors were input into a Random Forest Regressor (RF) model, the first time machine learning techniques have been applied to investigate BML thickness. The RF feature importance scores identified bottom depth, total internal tide energy dissipation, followed by slope, as the strongest predictors of BML thickness, revealing the importance of low-mode internal wave energy losses in this abyssal setting. Targeted and sustained observations near the seafloor at gateway regions of abyssal pathways are vital for understanding energy exchange that influences meridional overturning circulation. Our results highlight a regime where sustained low-mode internal tide energy loss, modulated by topographic slope and depth, governs the BML thickness in the abyssal Pacific. However, the rate at which BML thickness changes over time and the processes that cause these changes remain key unresolved factors.

## 1 Introduction

Nearly half the Pacific Ocean comprises abyssal zones that have experienced persistent warming in the past 30 years (Johnson and Purkey (2024)). The bottom mixed layer (BML), a well-mixed region directly above the seafloor in the abyssal ocean





(Armi and Millard Jr (1976); Lentz and Trowbridge (1991)) is a critical interface where turbulent mixing facilitates exchange between the deep ocean interior and the seafloor, influencing water-mass transformation and global circulation. Dynamics within the BML are affected by internal wave activity (Zulberti et al. (2022); Holmes et al. (2016)), and near-boundary turbulence (Lentz and Trowbridge (1991); van Haren et al. (2024)), impacting diapycnal mixing, deep-sea food web connectivity,

and heat transport (Jayne et al. (2004)). Additionally, the BML region may contribute to abyssal mixing, as the interplay of turbulent processes through internal tides and stratification here facilitates diapycnal mixing (Kunze et al. (2012)), thereby helping to drive the meridional overturning circulation (MOC) (Wunsch and Ferrari (2004); de Lavergne et al. (2017); Ferrari et al. (2016)). MOC is sustained by the abyssal flow of North Atlantic Deep Water (NADW) and Antarctic Bottom Water (AABW), which transport dense water masses equatorward and poleward from their formation regions through the deep ocean. The path-

ways involved in the MOC have been broadly identified, with general consensus on their origins, particularly in the ventilation regions adjacent to Antarctica (AABW) and the Labrador Sea (NADW) (Ferrari et al. (2016); Talley (2013)). While the broad pathways of these abyssal waters are accepted, the detailed mechanisms by which they return to the surface through diapycnal mixing and upwelling remain active areas of research (Marshall and Speer (2012); van Haren et al. (2024); Wynne-Cattanach et al. (2024); de Lavergne et al. (2017); Drake et al. (2022)). Regions of intense abyssal mixing over mid-ocean ridges and

narrow inter-basin channels are key sites where stratification governs regional variability in the BML, current pathways and internal tide generation, connecting closely to abyssal pathways, while seafloor topography remains the primary control on BML thickness at the global scale (Gula et al. (2016)). However, the relative importance between topography, its spatial scales, and the dynamic processes within ocean basins dictating the BML thickness remains unclear (Weatherly and Martin (1978); de Lavergne et al. (2017)).

The strength of deep MOC in the Pacific Ocean has been historically underestimated due to a lack of data and its complex topographies making simulations more challenging (Kawabe and Fujio (2010); Oka and Niwa (2013)). Broadly, AABW enters the Pacific Ocean along the eastern side of the Tonga-Kermadec Ridge (Chandler et al. (2024)), then narrows through the Samoan Passage (Alford et al. (2013)) before bifurcating to the west and north towards the Japan Trench (Kawabe and Fujio (2010)) (Figure 1). North of the Samoan Passage, there is also bottom water transport to the east and south of the Hawaiian

Ridge. Around the Hawaiian Ridge, energetic baroclinic tides are generated over the rough seafloor, contributing to distinct differences in the eastern and western regions of the Pacific Ocean, with larger dissipation in the western Pacific (Hautala (2018); Alford et al. (2007)). AABW transforms into North Pacific Deep Water (NPDW) while reaching the North Pacific. It is then further transformed through deep ocean mixing while travelling south and reinforcing subsurface stratification and linking to deep convection in the Southern Ocean (Tatebe et al. (2018)).

While rough and variable topography can modulate the BML thickness (e.g. fracture zones (Thurnherr et al. (2020)) and seamounts (Mashayek et al. (2024)), regions of the seafloor with broadly similar depths or geomorphology can nevertheless exhibit vastly different BML thicknesses due to differences in ocean dynamics such as boundary currents or abyssal transformations (Drake et al. (2022); Holmes et al. (2018)). Along continental shelf regions, it is on the order of 40-70 m in the South China Sea (Liu et al. (2023)) and 5-15 m along the Northern California Shelf (Lentz and Trowbridge (1991)). The mean

BML thickness over different latitudes in the North Atlantic Basin has been reported as 30-60 m (Lozovatsky and Shapovalov





(2012)). Across the Drake Passage, it was found to be over 100 m, similar to Gulf Stream regions (Todd (2017)). These differences may also be due to methodology. For example, the region immediately north of the Puerto Rico Trench in the North Atlantic ( 21°N, 66° W) has a BML thickness reportedly ranging from 80-800 m (Figure 9 in Banyte et al. (2018)), 60-100 m (Figure 2b in Lozovatsky and Shapovalov (2012)) and 80 m (Figure 1 in Huang et al. (2019)) with the variation likely attributed
to different methodology and spatial interpolation.

The transfer of mass and momentum between the ocean interior and the seafloor occurs via the BML. Yet in most large-scale ocean circulation models, it is generally unrepresented. As a result, robust parameterizations of bottom boundary processes are essential (Legg et al. (2006); Fox-Kemper et al. (2019)). Munk (1966) initially proposed a vertically integrated, one-dimensional framework to estimate diapycnal mixing rates averaged across the ocean interior. However, it has been subse-
quently found that mixing in the bottom boundary is inherently three-dimensional, shaped by turbulent processes influenced by topography and internal wave dynamics (Kunze et al. (2012); Polzin et al. (2014); Wunsch (2023)). Simple bottom boundary layer parameterizations assume local, steady-state velocity shear and stratification relationships to simulate turbulent mixing and momentum transfer vertically, potentially neglecting variability in the BML thickness (Large et al. (1994)). Recent improvements have incorporated wave-driven turbulence and terrain-following schemes (Arbic et al. (2009)), some of which
include profiles of diffusivity and viscosity in their parameterization (Fox-Kemper et al. (2019)). Nevertheless, additional observations along the ocean's bottom boundary remain crucial, not only for validating models, but for resolving the spatial and temporal variability in the BML structure that underpins interior-seafloor exchange (de Lavergne et al. (2017); Ferrari et al. (2016); van Haren et al. (2024); Wynne-Cattanach et al. (2024)).

The BML thickness is most commonly defined as the thickness above the seabed at which a variable (typically conservative
temperature or density) deviates from the seafloor value by a specified threshold (also known as the 'threshold method'). While we refer to this as the bottom *mixed* layer, its thickness may reflect varied bottom boundary processes, not exclusively active mixing. It is inhomogeneous throughout the world's oceans, not only because of the varying depth and roughness of the seafloor but also because of the influence of differing oceanographic processes in each region. Different oceanographic conditions require varying thresholds to calculate the BML thickness. For instance, weakly stratified abyssal regions necessitate
small density thresholds (1 x $10^{-3}$ kg m$^{-3}$), while highly turbulent areas are better suited to larger threshold values (Figure 1 in Banyte et al. (2018)), or sensitivity in the threshold value may be directly related to instrument noise (Lentz and Trowbridge (1991)). To overcome sensitivity and subjectivity in threshold (or gradient) selections, approaches like the relative variance (Huang et al. (2018b)) and integrated methods (Huang et al. (2018a)) have been developed to provide more robust, non-arbitrary BML thickness estimates. The BML thickness serves as a useful proxy for characterising diapycnal upwelling (de Lavergne
et al. (2017); Thurnherr et al. (2020)), nutrient transfer (Hull et al. (2020)), sediment transport (Edge et al. (2021)) and the development of bottom boundary conditions and parameterizations in ocean models; therefore, the methods and outputs require diligent evaluation of their physical validity across spatial and temporal scales.

Our first objective is to evaluate BML thickness methodological approaches suitable for data-poor regions of the ocean. We focus on four methods: the threshold method, the threshold-gradient method, a relative variance method, and a split-and-merge
algorithm to derive the BML thickness and critique their use and relevance in an abyssal basin setting. The second objective is



to understand the BML variability across the central and eastern Pacific abyssal ocean, including assessment of the connection to internal tidal energy dissipation and bottom water pathways. To achieve these objectives, we collected surface-to-seafloor temperature-pressure profiles across the central and eastern Pacific Ocean and complemented them with publicly available repeat hydrographic datasets. As we show in the sections that follow, these data reveal new insights into the dynamics of the
BML and its connection to broader abyssal processes.

## 2  METHODS

The Trans-Pacific Transit (TPT) Expedition occurred over six individual legs, with the duration of each leg approximately 21 days, between June 2023 and January 2024 on *Research Vessel (RV) Dagon* (Jamieson et al. (2024)). The vessel covered $20°$ longitude and $30°$ latitude over the central and eastern abyssal Pacific Ocean, including the Molokai and Clarion Clipperton
fracture zones (Figure 1). Bathymetry and backscatter intensity data were acquired throughout the expedition using a hull-mounted Kongsberg EM124 multibeam echosounder. At each site, three autonomous landers were deployed in a roughly 2 km equilateral triangle, acquiring a total of 73 surface-to-seafloor profiles of temperature and pressure (at a frequency of 2Hz secured to the landers). These data are referred to here as "TPT profiles". Conductivity, temperature and depth (CTD) sensor profiles from repeat Global Ocean Ship-Based Hydrographic Investigations Program (GO-SHIP) hydrographic sections, P16
and P02, provided observations within the study regions along meridional and zonal transects, respectively (see locations in Figure 1). The GO-SHIP sections were used in two ways: first, they were used to complete Gaussian mixture modelling (GMM), which was applied to the TPT profiles to generate modelled practical salinity (SP) profiles (see Methods section on GMM); second, they provided additional locations to derive the bottom mixed layer (BML) thickness. The BML derivation, followed by a random forest regressor, was applied to the GMM-derived TPT profiles and GO-SHIP profiles as one dataset and
is detailed in the following sections.

### 2.1  Data collection

Three autonomous landers, *Magna, Omma* and *Cranch* included a baited camera system, niskin bottles and a CTD (conductivity, temperature, and depth) sensor measuring at 1 or 0.1Hz (SBE49 FastCAT, SeaBird Electronics, Bellevue, WA). The landers descended at an average speed of 0.8 ms$^{-1}$, spent up to 8 hours on the seafloor and then returned to the surface by releasing
their ballast weights via an acoustic modem. On legs two to six, there was the addition of a temperature (t) and pressure (p) logger (RBRduet|deep) mounted to the lander frame measuring at 2Hz with an accuracy of $\pm0.002°$ C and 0.05% full-scale respectively (RBR (2024)). Due to the consistent high-frequency measurements of the t-p sensor, we have used this data for consistency and employed GMM to the profiles (see the following section). Therefore, Leg one data was omitted from this study and a total of 69 profiles were suitable for this study. The exact locations of deployments are in Appendix A1.
GO-SHIP profiles were obtained through the CLIVAR and Carbon Hydrographic Data Office (CCHDO, https://cchdo.ucsd.edu/) for repeat hydrographic sections P02 and P16 which form part of the GO-SHIP program. These were voyage numbers: *31WT-TUNES_3*, *325020060213* and *33R0150410* for P16 and *49K6K9401_1*, *318M200406* and *318M20130321* for P02. Only





profiles that exceeded 2,000 m and reached within 40 m of the metadata bottom depth were used to calculate the BML thickness. Only one occupation along line P16 went north of 23° N, therefore north of this latitude was excluded for line P16. A gridded version of GO-SHIP dataset, GO-SHIP Easy Ocean provided by Katsumata et al. (2022), and available from https://zenodo.org/records/13315689 was used to produce the background neutral density, $\gamma_n$, and conservative temperature, $\theta$, for Figures 5 and 6.

### 2.2 Gaussian mixture modelling in Θ-$p$-SA space

Gaussian mixture modelling applications (GMM) can achieve unsupervised classification of the water column, identifying coherent patterns in the associated domains (Maze (2017)). Considering this throughout a water column and our region, using depths only over 3,000 m, classification of the water column can be done using Θ-$p$-SA space within a relatively small volume (Hjelmervik and Hjelmervik (2014)). For hydrographic profiles close in proximity, this space is even tighter (McDougall and Jackett (2007)). Therefore, the most recent GO-SHIP profiles in the study region were used to predict salinity, in this case, practical salinity (SP) from 2,500 m to the seafloor using GMM (Pedregosa et al. (2011a); Maze (2017)). GO-SHIP profiles, available on CCHDO and collected within the last 5 years and within 10° latitude and 10° longitude that exceeded 2,000 dbar were used to predict SP for each TPT temperature and pressure profile. Model selection used information-theory criteria, focusing on the covariance type and number of components in the model using the Gaussian Mixture *scikit-learn* Python package (Pedregosa et al. (2011a)). The maximum number of components was limited to 21. The covariance types were limited to each component having its own general covariance matrix or all components sharing the same general covariance matrix. The elbow method was used to determine the number of components in the model with a brief examination of the BIC value. If the modelled SP output was physically unstable, the next best option was chosen. A further detailed explanation and model details for each TPT profile are provided in Appendix A1.

### 2.3 BML thickness derivation

Several methods exist for determining the BML thickness, as with the surface mixed layer thickness. The threshold method (TH) uses the depth at which the difference in either Θ or $\sigma$ is less than a defined threshold value. These values range from 0.02°C (Lentz and Trowbridge (1991)), 0.001°C (Hogg et al. (19821)), 0.005°C (Lozovatsky and Shapovalov (2012)) to 6 x $10^{-4}$ kg m$^{-3}$ (Perlin et al. (2005)). We used a threshold value of 0.003°C for the region. This value was chosen as it provided the highest mean quality index (QI) for the BML thickness (1) for all the TPT profiles when comparing threshold values of 0.001, 0.002, 0.003, 0.004, and 0.005 (Appendix A2). A quality index was initially defined by Lorbacher et al. (2006) as a value between 0 and 1 capturing the conservative temperature variability in 1.5 times the BML thickness compared to the variability over the BML thickness. In equation form:

$$QI_{\text{BML}} = 1 - \frac{A1}{A2} = 1 - \frac{\sigma(\Theta_n - \langle\Theta\rangle)\big|_{(h_1, h_{\text{BML}})}}{\sigma(\Theta_n - \langle\Theta\rangle)\big|_{(h_1, 1.5 \times h_{\text{BML}})}} \tag{1}$$

where $\sigma()$ is the standard deviation from the vertical mean $\langle\rangle$ conservative temperature from $h_{\text{BML}}$.





The threshold-gradient method (GR) is also used (Banyte et al. (2018); Weatherly and Martin (1978)). We defined this method as the thickness at which $\delta\sigma_4/\delta z$ over 20 m intervals is less than a criterion of $1 \times 10^{-3}$ kg m$^{-3}$ (Banyte et al. (2018)), making the minimum BML thickness 10 m.

Several techniques have been put forward over the last decade, including a relative variance (RV) method (Huang et al. (2018b)), split-and-merge algorithms (Thomson and Fine (2003)) and an integrated method (Huang et al. (2018a)) that combines several methods together. We included the RV method and the Douglas-Peuchker (DP) algorithm method. The RV method relies on calculating the ratio between the standard deviation and the greatest variation of $\Theta$ or $\sigma$ above the seabed. The location where the least relative variance occurs is identified as the upper boundary of the BML. The RV method is available through the original research (Huang et al. (2018b)). The DP method is a split-and-merge technique that has been previously adopted to calculate the surface mixed layer (Thomson and Fine (2003)). The DP algorithm estimates a given profile by using a series of simplified line segments that represent large changes in slope or any abrupt changes in the profile. Therefore, the lowest part of the segment belongs to the BML. The DP algorithm is available within MATLAB and requires a $\epsilon$ value between 0 and 1 to determine the number of line segments (Ahmadzadeh (2024)). We included 0.002 and 0.008 as two possible DP methods (DP2 and DP8, respectively) (Appendix A2).

The integrated method put forward by Huang et al. (2018a) focuses on the use of multiple methods (TH, the curvature method (Lorbacher et al. (2006)), the maximum angle method (Chu and Fan (2011)) and RV (Huang et al. (2018b))), calculating the QI for each method, and then choosing the BML thickness with the highest QI (Huang et al. (2019)). Relying on QI-based selection of BML thickness from multiple methods produced highly variable results, even across nearby locations (within 3 km). That is not to say this variation may not be real, but visual inspection was still needed to assess accuracy, and the variation was not consistent with global maps from either Banyte et al. (2018) or Huang et al. (2019). Therefore, unlike Huang's global integrated approach, the single threshold method that produced consistent results was more appropriate for our regional study, avoiding unnecessary and possibly unreal variability that ultimately required manual validation.

The suitability of the threshold value, despite sometimes having a lower QI, is shown in Figure 2 at different locations with additional annotation in Figure 2(c). At times, the QI did capture the BML thickness values that were unusable (e.g. small RV QI values in Figure 2f). However, on the upper scale, the highest QI value could provide an unlikely, significantly larger BML thickness than where the density gradient approached zero and appeared visually correct. For example, in Figure 2c for the profile on the left side, the highest QI was 0.78 and far from the visual BML thickness height. Considering all TPT and GO-SHIP profiles, consistency in the average values of each method (Figure 4) and their performance when assessed visually (Figure 2), using the threshold method consistently over the whole dataset, provided the most reasonable result.

## 2.4 Random Forest Regression

We considered bathymetric parameters (terrain roughness index (TRI) and slope) and dynamic parameters (internal tide energy dissipation and depth) within a Random Forest Regressor (RF) to disentangle patterns in the BML thickness. Machine learning techniques have been used to estimate the surface mixed layer depth, however this the first time it has been applied to the BML (Imchen et al. (2025); Foster et al. (2021)). The RF machine learning technique, included as the *RandomForestRegressor*





scikit package in Python (Pedregosa et al. (2011b)) is a ensemble machine learning estimator that combines the outputs of multiple decision trees to increase predictive accuracy and control overfitting. Each decision tree is trained on the dataset using

a bootstrap aggregation technique, and the final prediction is obtained by averaging the outputs for regression (Carvalho et al. (2019); Imchen et al. (2025)). We randomly selected 80% of the dataset to build each tree and the remaining 20% of the dataset to test the model using *train_test_split* within scikit (Pedregosa et al. (2011b)). We modified the number of trees from 100 (default value) to 500 and 1,000 for sensitivity testing (1) with the random state of the *train_test_split* at 42, 0 or 1. The number of trees did not add significant computing time or signficantly alter the results, therefore we maximised the number of trees

remained at n = 1,000 to further test different random subsets of the data with *train_test_split* then changed between 42, and 0 to 7. The random state was kept at 42 for the RF for all itterations.

As discussed in the Data Collection section, internal tide energy dissipation (Wm$^{-2}$) for all (M2, S2 and K1) tidal constituents broken down into dissipation processes; low-mode wave-wave interactions, low-mode scattering by small-scale topography, low-mode interaction with critical slopes, low-mode shoaling and local dissipation of high modes was accessed

through the paper by Lavergne et al. (2019). These variables, alongside slope and TRI were chosen based on their accessibility and relevance to BML thickness Ruan et al. (2017); Gula et al. (2016); Liu et al. (2023). Depth is a known contributor to the BML thickness (Huang et al. (2019); Lozovatsky and Shapovalov (2012)) as are bathymetric variables of slope and TRI (Armi and Millard Jr (1976); Wunsch (1970); Polzin and McDougall (2022)). The relative contribution of tidal dissipation mechanisms near the seafloor, therefore influencing the BML thickness has been discussed in literature (Lavergne et al. (2019)).

When internal waves hit the seafloor, they lose energy through either scattering off small rough spots and losing energy, or reflecting or shoaling off topographic features, depending on their shape and height (St Laurent et al. (2001); Müller and Xu (1992). These processes, along with others that are not as well understood, like wave capture and scattering by mesoscale eddies Bühler and McIntyre (2005); Polzin (2008); Mathur et al. (2014), can speed up the dissipation of tides and change the thickness of the bottom mixed layer. A grid point from each of the dissipation parameters was assigned to each GO-SHIP and

TPT point using cKDTree in scipy for nearest-neighbor lookup (Virtanen et al. (2020). This method constructs a binary space-partitioning tree applying axis-aligned hyperrectangles via the sliding midpoint rule (Maneewongvatana and Mount (1999)). This provides efficient nearest-neighbor queries by recursively improving the search space across coordinate axes to determine the nearest latitude and longitude grid point to the GO-SHIP and TPT observations.

Bathymetric variables (TRI, and slope) were compiled from the latest GEBCO 2024 Grid, including the standard deviation

(GEBCO Compilation Group (2024)). TRI and slope were calculated using the ArcGIS Geomorphometry & Gradient Metrics toolbox with a neighbourhood of 9 x 9 cells (Evans and A (2014)). TRI is a useful derivative of bathymetric and topographic datasets in order to enable quantification of the spatial heterogeneity of the surface under investigation (Riley et al. (1999)). The TRI metric can be a valuable analytical tool for understanding the effect of landscape on processes, geomorphological evolution, and for habitat mapping and modeling regimes. For the extent of the study region (Figure 1) the slope and TRI

were calculated at buffer zones of 25, 50, 100 and 200 km (Figure A4). At each GO-SHIP and TPT data point, the TRI was extracted to assess the variation for different buffer zones. The RF was completed with the 50 km buffer, as the resolution



for the dissipation values was 50 km. The depth for the GO-SHIP sites was obtained from the datasets as the 'bottom depth' variable available in the datasets.

The GO-SHIP observations included multiple occupations as detailed in the Data collection section above. In some loca-
225 tions the exact latitude and longitude was covered in multiple years, although this is not spatially consistent throughout the observations. The profiles over the different occupations provide different BML thicknesses, however it is impossible to deduce the reasoning behind the differences at these yearly timescales as we know the BML thickness may change within a matter of hours (Weatherly and Martin (1978); Chen et al. (2023)). For this reason, we clustered the GO-SHIP observations within the RF using *dbscan* in scikit (Pedregosa et al. (2011b)). Geographic clustering was performed by converting the GO-SHIP
latitude and longitude coordinates to radians and applying *dbscan* with a haversine metric and a 3 km neighborhood radius to group nearby data points. For consistency between the TPT and GO-SHIP sites, TPT sites within a 3 km radius of one another (each leg and site) were averaged and GO-SHIP sites were averaged based on the 3 km neighborhood radius from *dbscan*.

# 3 RESULTS

## 3.1 BML thickness

The average and median thickness of the BML using the TH method for all data points in the abyssal study region was 240 m and 176 m respectively, and the standard deviation was 200 m (Figure 3). The BML was inhomogeneous over the region, its thickness decreasing around continental slope regions approaching Mexico and the southern part of Hawaii. Between 15°S and 2°N the BML was below 200 m. There was a distinct change between 2°N and 15°N where the BML approximately doubled and reached a maximum of 799 m crossing the Clarion Fracture Zone before decreasing to below 90 m south of Hawaii.
Along the zonal section of P02, the BML exhibited an approximately 50% increase between 135 to 130°W and decreased to approximately 100 m on approaching the continental slope. The TPT expedition data indicated generally similar patterns as the repeat hydrographic sections. These patterns excluded Leg 4 Site 7, where the BML was the largest of the TPT sites (Figure 3).

To provide insight into the efficacy of the derivation methods, we calculated a QI (Lorbacher et al. (2006)) for each profile
and its five possible BML thickness values (Figure 4). Visual inspection of the BML thickness estimates and their associated QI indicated that a higher QI and lower standard deviation do not always provide confidence in the BML value. TPT profiles within 3 km of each other in Figure 2e had a higher QI for the GR and DP8 methods; however, the methods estimated very different BML thicknesses for very similar profiles. In contrast, the TH method, with lower QI values, was consistent among the profiles and appeared to capture the position of profile change under contrasting abyssal conditions sufficiently. For example,
across all Figures 2e - h the TH BML thicknesses were in close proximity to one another. Additionally, the TH thickness corresponded with the visually identifiable thickness, despite the GR values being close in value to each other and of a high QI (Figure 2c). We therefore chose the TH method for all BML thickness values going forward due to its dependable performance when applied to both TPT and GO-SHIP profiles over different regions in the study area.





## 3.2 Spatial variability

On meridional transect P16 between 4 and 16°N, depths are over 5000 m, and the BML thickness was at its greatest along the transect (Figure 5). The transition regions at 4°N and 16°N appear to have the widest variation in BML over the different occupations. The TRI was low over the majority of this region, with increases near the Hawaiian Islands, over the Boudeuse Ridge (10 °S) and other prominent seafloor features visible in the bathymetric data (Figure 7b and c). Similarly, the TRI reached a maximum across transect P02 when it crossed the Murray Fracture Zone and the Moonless Mountains before increasing 260 towards the North American continental slope (see Figure 1 for locations). TPT sites close to the major fracture zones and seamount chains had higher TRI and slope values over the 50 km buffer range and exhibited patterns connecting them to the P16 and P02 lines spatially. The P02 transect had a higher BML thickness ($298 \pm 170$ m) compared to P16 ($175 \pm 157$ m) (Figure 5 and Figure 6). Similar to P16, the sections of changes in BML thickness along P02 (> 500 m from 136 to 129 °W) broadly intersected with larger differences in BML thicknesses over the different occupations (Figure 6).

## 265 3.3 Random Forest Regression

The TPT and GO-SHIP profiles were analysed together as part of a RF. As described in the Methods section, GO-SHIP data points within 3 km of one another were averaged together as a mean value to remove instances of temporal variability at unknown time scales. After the averaging, the number of GO-SHIP data points reduced from 335 to 301. The number of TPT data points was 29; therefore, a total of 330 points were used for the analysis. For each point, we extracted values of bottom 270 depth (m), slope, TRI, and internal tide energy dissipation values of low-mode wave-wave interactions, low-mode critical slope, low-mode scattering, low-mode shoaling, high-mode ($\geq 6$) local dissipation and total internal energy dissipation which is a sum of all the losses from the five processes (in W m$^{-2}$) (Figure 7). The dissipation parameters are defined in depth by Lavergne et al. (2019).

The feature with the highest importance score (∼0.4) across all iterations of the RF was the bottom depth. For each variation 275 of *train test split* sample data (i.e. *random_state* = 42, 0 or 1) chosen to train the RF, the same features with the highest importance were in the top 3. In order, these were the bottom depth, total dissipation and slope. The number of iterations (n = 1000, 500 or 100) and the three most commonly used *train test split* values (*random_state* = 42, 0 or 1) were sensitivity tested due to the relatively small number of data points. Reducing the predictor variables to include only the top 5 features, ranked by importance, increased the correlation coefficient, $r^2$, by ∼0.02, regardless of the number of iterations or the *random_state* value 280 (Table 1). Similarly, the root mean squared error (RMSE) and mean average error (MAE) reduced an insignificant amount (∼2 m) when including only the top 5 features (Table 1). The results from n = 1,000, 500 or 100 had comparable feature importance scores; therefore, only the results from n = 1,000, all features and additional *train test split* values of *random_state* = 42, 0-7 were run and are shown in Figures 8 and 9. If we were to keep the *random_state* value as empty, which is the default, the sensitivity in altering the number of iterations would not be effectively tested. *Train test* values of *random_state* = 42 and 0-7 285 were completed for n = 1000 with the $r^2$ and RMSE displayed in Figure 9. The nine sets of the RF residual model outputs are





shown in Figure 9. The $r^2$ was between a minimum of 0.53 and a maximum of 0.77 and the RMSE had a minimum of 87.1 and a maximum of 127.

The different *random_state* values changed the ranking order of the feature importance scores, however the same features were within the top five, with the bottom depth always the highest. The slope and TRI are intrinsically linked due to their
calculation from the same bathymetric dataset, with slope quantifying the local gradient over a 50 km radius and TRI capturing the variability within a neighborhood, averaged over a 50 km radius aligning with the spatial resolution the dissipation values, a 0.5° grid size. A surface may be steep but smooth, or flat yet jagged, drawing not always a strong correlation between the two. For example, there are sharp changes latitudinally, however, both TRI and slope are high and more gradual along the P02 line as part of the broad sloping region from the continental slope of Mexico to the center of the Pacific Ocean (Figure
7b and c). The western end of the P02 line has higher internal tidal dissipation compared to the more eastern half due to the presence of the Hawaiian Islands (Kelly et al. (2010)). The full spatial extent of the dissipation parameters, not just at our data points, at 0.5 ° resolution are displayed and explained in Lavergne et al. (2019). Overall, the internal tidal dissipation for each low-mode process (Figure 7 e-h), and the total dissipation (Figure 7d), is highest between Hawaii and just north of the equator, intersecting with the region of higher BML along P16 aside from 15 - 20° N next to the Hawaiian Islands, where the BML
decreases. This decrease does overlap with a slight decrease in bottom depth and total dissipation, presumably enhancing the importance of total dissipation and low mode (wave-wave) dissipation in the RF.

## 4  DISCUSSION

Density profiles over the central and eastern Pacific Ocean provide an inhomogeneous outlook of BML thickness variations at abyssal depths across plains and topographic features. Basin-scale expeditions such as the TPT voyages are frequently
multidisciplinary in scope, with competing demands on vessel time. Incorporation of these profiles with the repeat GO-SHIP profiles provides increased understanding of the BML. Using RF methods, we found that bottom depth, total internal tide dissipation and slope are the highest-performing features to predict the BML thickness in this region.

In the central and eastern Pacific abyssal ocean, the thickness of the BML was inhomogeneous with an average value of $226 \pm 172$ m. The BML was calculated using the ($\Theta$) profiles through the threshold method (0.003°C) for the study region.
There is no accepted standard methodology for calculating the BML thickness. It often depends on user-defined numerical values within those methods and depends on the region of interest. The integrated method proposed by Huang et al. (2018a), was used to calculate the BML depth globally, providing an average Pacific Ocean BML thickness of 64 m (Huang et al. (2019)). We found that for our abyssal ocean context, using an integrated approach that combines multiple methods and calculates a QI to get the highest 'quality' BML thickness generated spurious results for profiles within three kilometres of
one another, making it difficult to compare BMLs estimated with different methods. Although the QI was calculated, visual interpretation was necessary to confirm the results, mirroring the approach taken within the integrated method where visual identification was still needed (Huang et al. (2019)). The variability in BML thickness is not unexpected given the variation in topographic features across the region, likely changes in friction velocity, and a wide longitudinal and latitudinal range





(Weatherly and Martin (1978); Kunze et al. (2012)). Profiles from the TPT Expedition broadly followed the same spatial

patterns as those from the hydrographic sections and show similar spatial variations in BML thickness as Banyte et al. (2018).

In all instances, the RF identified the bottom depth, slope, total internal wave energy dissipation, TRI and low-mode wave-wave interactions as the most important predictors of the BML thickness in this Pacific Ocean abyssal setting. These results are physically intuitive, with the bottom depth constraining the maximum possible BML thickness, background stratification and the vertical extent available for turbulent mixing, which is consistent with past research (Laurent and Garrett (2002); Liu

et al. (2023); Lozovatsky and Shapovalov (2012)). The total internal wave energy dissipation value aggregates all internal tide dissipation mechanisms that drive turbulence (Lavergne et al. (2019)). On abyssal plains, where local topographic features are sparse, a substantial amount of this energy would likely originate remotely and dissipate gradually through sustained mixing events (Nikurashin and Legg (2011)). Therefore, the inclusion of low-mode wave-wave interactions as a predictor is especially significant. This variable refers to nonlinear energy transfers among long-wavelength internal tides. In regions with high low-

mode wave-wave dissipation, this could lead to persistent near-bottom mixing that expands the BML thickness. This suggests that the BML thickness on the abyssal plain is of remote and sustained forcing origin, rather than high-mode breaking events (Lavergne et al. (2019); Melet et al. (2013)). Although the study region lies predominantly within the abyssal plain, the terrain is not uniform. As shown by Harris et al. (2014) and Figures 7b and c, the region is interspersed with multiple features of abyssal plains, abyssal hills, and seamounts, creating heterogeneity in the TRI and slope. The observations north of Hawaii

highlight where higher TRI and slope intersect with the smallest values of total internal tide dissipation and low-mode wave-wave interactions.

The TRI captures local bathymetric complexity at 50 km scales, which enhances bottom drag and internal tide scattering, even where mean slopes may be weak, supporting thicker BMLs by maintaining sustained and patchy mixing close to the boundary layer (Nikurashin and Ferrari (2011); Nash et al. (2007)). The slope of the topography within our study region is

primarily of a subcritical regime; therefore, internal tides will refract and reflect weakly, allowing for persistent low-mode energy to mix over broader regions, rather than localized mixing. Between 4 and 15° N there is a small region where the low-mode critical slope dissipation increases (Figure 7g), and the BML thickness is large, suggesting the critical slope may be more important here. Nested within the same region is the highest total dissipation of the study region (4 - 7° N) where there is high local high-mode dissipation, low-mode wave-wave interactions and low-mode critical slope dissipation. The TRI and

slope are small over this region, culminating in the BML thickness being slightly above the average. Similar connections to the slope and the TRI have been identified in the North Atlantic Ocean and the South China Sea (Lozovatsky and Shapovalov (2012); Liu et al. (2023)).

Our results have highlighted differences in what factors drive the BML. Despite limited and sparsely located data points, it is clear that the BML thickness is a culmination of processes, both local and remote. The limited and spatially inconsistent data

points meant we were unable to further the model to predict the BML without the RMSE at times equating to the predicted BML thickness. Despite the three highest importance features remaining consistent, the nine iterations of *random_state* values do not visually provide a clear picture of regions that are consistently lower or higher performing than others. In addition, the reasonably high variation in r$^2$ and RMSE values suggests that more observations are required for there to be less sensitivity





in the results to which random selection of the data is chosen to train the model. The RF herein should be used to understand
drivers but cannot be used predictively, which would require additional data points. However, in regions where there is a more
dense and equal spread of CTD profiles, the publicly available datasets from Lavergne et al. (2019) and GMRT bathymetry
(Ryan et al. (2009)) should be considered for usable predictive relationships. Prediction of the BML thickness was not in the
scope of this study; however, we have shown the usefulness of publicly available datasets. Predictive relationships of the BML
thickness would be useful for identifying regions of interest for internal wave-driven mixing at the ocean's bottom boundary,
hydrodynamic model parameterisations and disentangling spatiotemporal variability in BML thickness within a given region.

At around 18°N, bottom water passes through the Horizon Passage (170° W) and flows around Hawaii (Lukas et al. (2001);
Fuhr et al. (2021); Kato and Kawabe (2009)), intersecting where the BML was small and there was increased stratification in the
water column above. This can be demonstrated by Θ-SA profiles with increased fractions of North Pacific Intermediate Water
(NPIW) from 16-19° N (Figure 10a) and more saline and cooler water at the seafloor within the BML, aligning with Antarctic
Bottom Water (AABW) properties (Figure 10b). While the complete profiles between 0-2 and 10-15° N displayed visually
similar water mass characteristics, the properties of the BML were distinct (X marks in Figure 10) with the equatorial seafloor
BML fresher and warmer, indicating NPDW, compared to 10-15° N closer to the properties of AABW, and 16-19° N the most
saline and coolest (Fuhr et al. (2021)). This region of water mass and inter-basin exchange highlights the difference between
stratified regions of bottom water pathways (Figure 1) compared to low ocean interior stratification south of this region (curved
Θ-SA in Figure 10b, red and blue) and less variation in Θ-SA space (Figure 10a) (McDougall and Jackett (2007); Hautala
(2018); Kawabe and Fujio (2010)). In essence, a more strongly stratified ocean interior likely suppresses mixing by reducing
the turbulent diffusivity, even in regions where the turbulent kinetic energy dissipation may be high. Therefore, the buoyancy
gradient remains difficult to overcome, resulting in a thinner BML (Weatherly and Martin (1978)).

Consistent with previous analyses (Liu et al. (2023); Lozovatsky and Shapovalov (2012); Chen et al. (2023)), there are
multiple processes influencing the BML thickness at abyssal basin scales. While the RF provides a quantitative approach to
dissect the variations in BML thickness based on the features, the profiles are a single snapshot of the water column at that point
in time. As exemplified by Chen et al. (2023) in the Clarion-Clipperton Fracture Zone, they were unable to define the diffusion
processes of the suspended sediment within the BML, as additional short-term processes such as internal gravity waves were
highlighted as likely influencing the results. In our case, the TRI and slope are single values over a 50 km buffer region, not
including proximity and direction from features such as the Hawaiian Ridge, which may influence the formation of the BML
and the water column in different ways; hence the inclusion of internal tidal dissipation from de Lavergne (Zaron (2019);
Finnigan et al. (2002)). Considering the broader consequences of BML dynamics for deep ocean mixing and overturning
circulation, temporal variability in the BML over abyssal depths should be considered in future studies. For example, a mooring
configuration both within the BML thickness and above, in a region of increased internal tide energy dissipation south of
Hawaii and then at a similar depth to the north-east of Hawaii where the BML thickness is higher and dissipation is lower,
while intersecting with a region of water mass transport. These locations transition from flat abyssal plains to the Hawaiian
Islands, each with distinct BML patterns and drivers across ∼six degrees of latitude. Increased observations of both direct





mixing and far-field or wave-wave energy dynamics are required in this relatively dynamic, yet undersampled region of the abyssal Pacific Ocean.

## 5   Conclusions

The BML is crucial for understanding diapycnal transport, which causes significant upward movement of deep-sea waters (Mc-Dougall and Ferrari (2017); Ferrari et al. (2016)) in the undersampled abyssal ocean. This research highlights the importance of abyssal seafloor regions, which are not typically categorized as dynamic, shifting in space and time. Through the application of four BML detection methods, we find that the commonly used threshold method provides the most consistent and interpretable estimates of BML thickness across large spatial scales. However, we have highlighted the necessity to test each method-specific parameter. We also show that GMM offers a useful approach for predicting essential ocean variables, such as salinity here, using publicly available data. The RF revealed BML thickness variation related primarily to bottom depth, followed by total internal tide energy dissipation and topographic slope.

   Global studies of the BML using multiple methods seldom focus on the potential of variability over time (Huang et al. (2019)), while others aim to use a small range in time to comprehend processes such as sediment dispersal within the BML (Liu et al. (2023)) reaching the conclusion of significant temporal variability long noted in literature (Greenewalt and Gordon (1978)). The relative contributions of the mechanisms that control the BML thickness across the abyssal ocean and basin boundaries requires further investigation through increased continuous observations and modelling efforts with reduced interpolation. The role of abyssal circulation pathways and internal tide driven mixing is at the forefront of current research (e.g. Wynne-Cattanach et al. (2024); van Haren et al. (2024)), within which the formation of the BML forms a key component of the processes. Therefore, the present study highlights and encourages sustained observations of abyssal regions over the bottom boundary and ocean interior above. Such observations are particularly important around rough topography, specifically in the central and eastern Pacific, where the abyssal ocean is frequently overlooked. At present, the temporal scales of BML variability remain poorly understood. Determining these scales is essential for characterising how the BML is mediated by abyssal water-mass transformation and circulation.

*Data availability.*   GO-SHIP profiles were obtained through the CLIVAR and Carbon Hydrographic Data Office (CCHDO, https://cchdo.ucsd.edu/) for cruise numbers: 31WTTUNES_3, 325020060213, 33R0150410, 49K6K9401_1, 318M200406 and 318M20130321. The gridded GO-SHIP product from Katsumata et al. (2022) was also accessed, used within figures and available on Zenodo at *zenodo.org/records/13315689*. The temperature-pressure sensor observations collected over the Trans-Pacific Transit Expedition on board *RV Dagon* are currently available on Zenodo at *zenodo.org/records/15536316*. The global maps of internal tide generation and dissipation as outputs from Lavergne et al. (2019) are available from SEANOE at *seanoe.org/data/00469/58105/*



**Appendix A: Gaussian mixture modelling of TPT salinity profiles**

We applied Gaussian mixture modelling to achieve a modelled representation of the practical salinity (*SP*) from 2,500m to the seafloor at each voyage deployment location (Table A1). Such unsupervised classifications have been completed for CTD profiles and Argo floats (Ye and Zhou (2025); Zhang et al. (2023)). All GO-SHIP profiles deeper than 5000 m and within 10° latitude and 10° longitude from the voyage site were used in scikit-learn package GaussianMixture (GMM) (Pedregosa et al. (2011a)). If there were 2 or less GO-SHIP profiles within the bounding box, it was expanded to 15° latitude and 15° longitude, otherwise the site was excluded. Mixture models can be viewed as an extension of $k$-means clustering that integrates information regarding the covariance structure of the data alongside the centres of the latent Gaussian distributions. They are a probabilistic framework that assumes all data points are derived from a combination of a finite set of Gaussian distributions with unspecified parameters. Options within the package that were altered to get the optimal GMM model of SP were:

- $covariance\_type$: tied or full, default is full.

- $N\_components$: $1 - 21$, number of mixture components.

- $random\_state$: 42, controls the generation of random samples.

The rest of the parameters were kept as default values. Each set of GO-SHIP data for the associated TPT voyage site was iterated through each covariance type for each number of components. The elbow method was then used to choose the number of components, whereby the increase in the number of components does not equate to an increase in the model performance ($AIC/BIC$).

**Appendix B: Sensitivity analysis for threshold value (TH method) and $\varepsilon$ value (DP method)**

We used the collected profiles of temperature and pressure to test the optimal threshold value to use for the BML thickness derivation. We used the Quality Index methodology (Equation 1 in the main text) to choose the appropriate threshold value based on Lorbacher et al. (2006) as it was being applied to the same method. The conservative temperature of $0.003°$ C provided the highest mean QI for both the TPT voyage dataset (Figure A1) and the GO-SHIP repeat hydrographic sections of P02 and P16 (Figure A2).

The Douglas-Peucker split-and-merge algorithm (Ahmadzadeh (2024)) reduces the number of points in a curve, approximating it by a series of points. An $\varepsilon$ value between 0 and 1 is required to specify the similarity between the curve and the points, i.e. the smaller the epsilon, the more similar the curve. We tested three TPT Expedition profiles with $\varepsilon$ values between 0.001 and 0.01 (Figure S3 for site $TP2\_OM3\_5400$). We chose $\varepsilon = 0.002$ and $\varepsilon = 0.008$ as the two options with the most variability in results to calculate the BML height. This gave us methods DP02 and DP08 respectively.

*Author contributions.* JK: Data curation, Formal Analysis, Investigation, Methodology, Writing - original draft, Writing — reviewing and editing. DH: Data curation, Writing - reviewing and editing. NJ: Methodology, Investigation, Writing - reviewing and editing. JO: Method-



ology, Writing - reviewing and editing. TS: Methodology, Writing - reviewing and editing. TB: Chief Scientist on Leg 3. HS: Chief Scientist on Leg 6. AJ: Chief Scientist on Leg 1, 2 and 5, Funding acquisition. All authors reviewed the manuscript.

*Competing interests.* The authors declare there are no competing interests

*Acknowledgements.* The authors would like to thank Inkfish LLC for their continuing support and logistics. We thank Captain Stuart Buckle, Captain Alan Dankool, Captain Jim Wales, Captain Ali Benarabi, the crew and the company onboard R/V Dagon for their crucial role in the successful completion of the Trans-Pacific Transit Expedition 2023-24. Furthermore, we thank the hydrographic surveyors Jaya Roperez, Erin Heffron and Tion Uriam; and the science, submersible, and lander teams (Ryan Beecroft, Murray Blom, Bruce Brandt, Chris Corcos, Megan Cundy, Samuel Dews, Shane Eigler, Paul Fairclough, Brett Gonzalez, Andrew Henderson, Jeff Huck, Reuben Kent, Catriona Macdonald, Tim
MacDonald, Alfredo Marchio, Shane Muhl, Georgia Nester, Gary Ogden, Alan Scott, Sarah Searson, Luke Siebermaier, Melanie Stott, Kate Wawatai, Brett Wilkins, Eddo Van Kolck, and Jennifer Wainwright). Taimoor Sohail acknowledges funding from the Australian Research Council Discovery Project DP240101274.



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





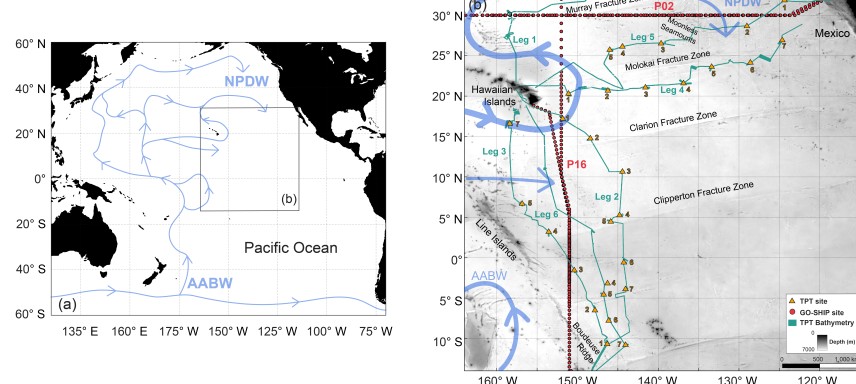

**Figure 1.** (a) Bottom water circulation pathways through the Pacific Ocean based on existing research (Kawabe and Fujio (2010); Oka and Niwa (2013); Hautala (2018)) with (b) insert as the extent of the study region. AABW = Antarctic Bottom Water, NPDW = North Pacfic Deep Water (b) Study region boundary, locations, and features, including a regional bathymetric grid. Orange triangles are the Trans-Pacific Transit Expedition deployment locations with numbers as the site number within the associated leg. The R/V Dagon multibeam echosounder coverage is displayed in green. Note that Leg 1 is not used in this analysis. The GO-SHIP repeat hydrographic lines and deployment locations are marked with red circles (P16 and P02). Blue arrows are bottom water circulation pathways adapted from previous studies. Background regional bathymetry is from the Global Multi-Resolution Topography (GMRT) Synthesis (Ryan et al. (2009)) Released CC BY 4.0 Deep | Attribution 4.0 International | Creative Commons.





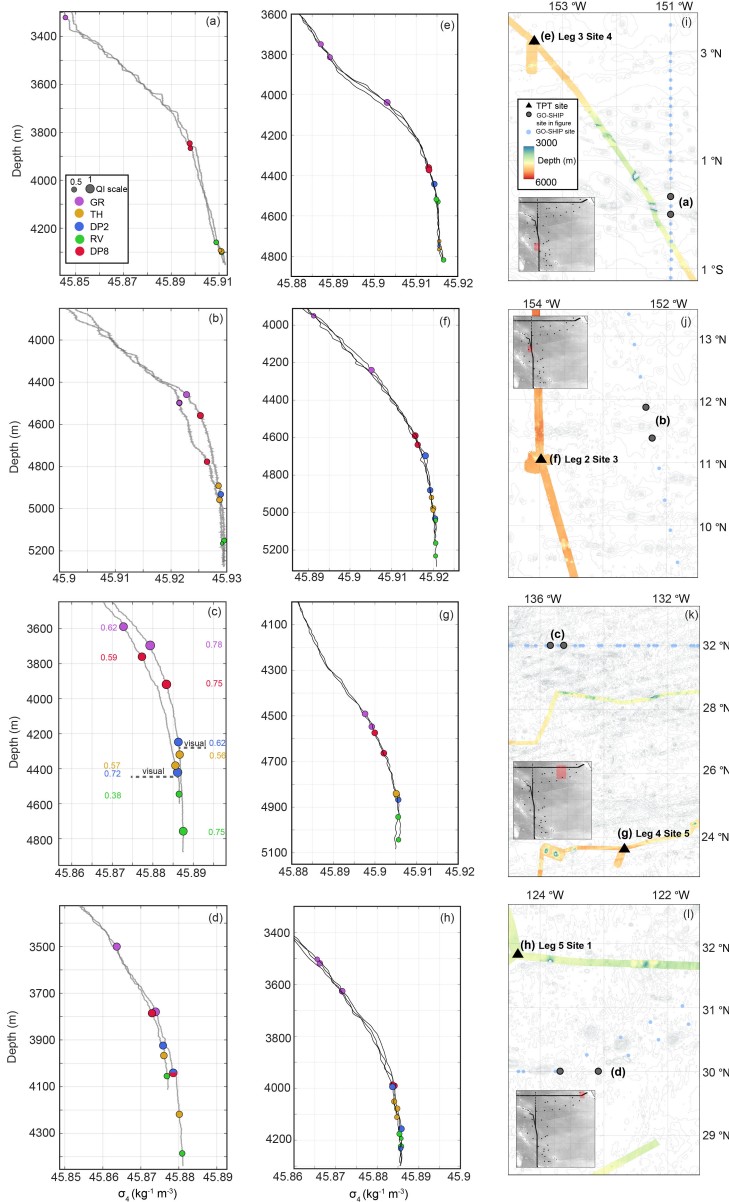

**Figure 2.** Example profiles of $\sigma_4$ and BML thickness outputs from (a) to (h). Color and size of the marker correspond to the method and the quality index (QI) of the BML (a). The map and inserts in (i) to (l) detail the locations of the profiles shown in the respective subplot. Grey map shows the study region with GEBCO bathymetry and sites (key in (i); see Figure 1 for more detail), the red region the extent of the figure and black points indicating all profiles used. Blue circles are GO-SHIP with light blue indicating profiles used but not displayed, black triangles are TPT locations. (a) P16 GO-SHIP profiles in 2002 between $0°$ and $0.5°$N and (b) in 2015 between $12°$ and $13°$ N, (c) P02 GO-SHIP profiles in 2022 between $135°$ and $136°$ W and (d) in 2022 between $124°$ and $123°$ W. (e) TPT voyage Leg 3 Site 4, (f) Leg 2 Site 3, (g) Leg 4 Site 5, and (h) Leg 5 Site 1. Figure (c) includes a line of visual interpretation and the exact values of the quality index, as also indicated by the marker size at the BML thickness for each method.





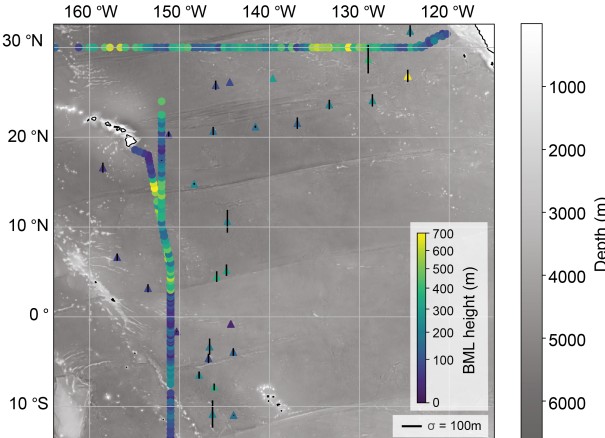

**Figure 3.** Bottom mixed layer (BML) thickness (m) derived from the threshold method (TH). It is calculated using the TPT Expedition profiles (triangles) and GO-SHIP profiles (circles). The TPT profiles are within 3 km of each other and therefore a standard deviation is included.





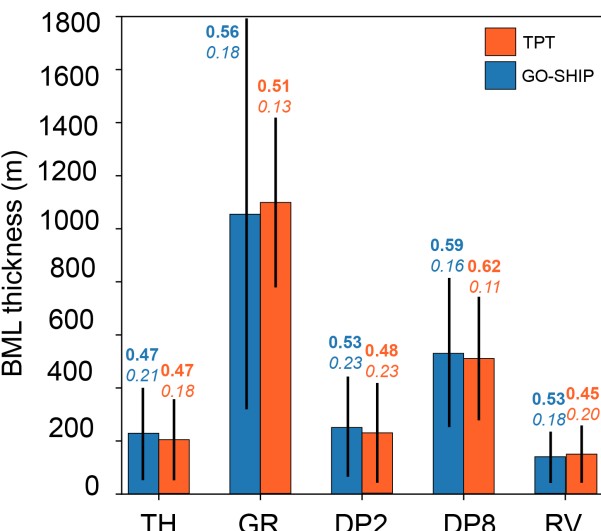

**Figure 4.** Average bottom mixed layer (BML) thickness (m) for the threshold method (TH), gradient method (GR), Douglas-Peuker method using an $\epsilon$ of 0.002 (DP2), Douglas-Peuker method using an $\epsilon$ of 0.080 (DP8) and the relative variance method (RV). The BML thickness is calculated using the TPT Expedition profiles (orange) and GO-SHIP profiles (blue) with the length of the line indicating the range of the thickness. Bold values above the bars are the mean quality index (QI) and the italicised values are the standard deviation of the QI.





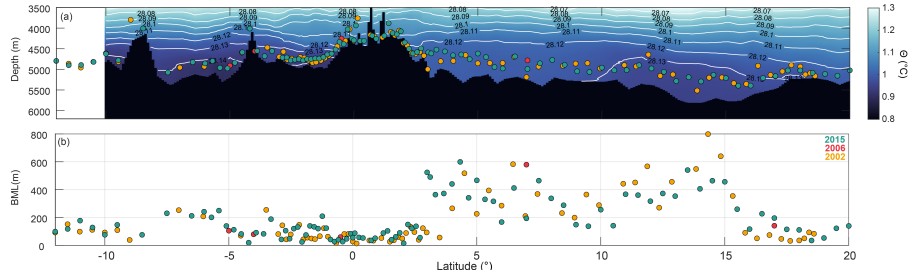

**Figure 5.** The P16 repeat hydrographic line nominally along $150°$ W with (a) conservative temperature ($\Theta$) with neutral density ($\gamma_n$) as the white contours and the BML thickness for 2015 (green), 2006 (red), and 2002 (orange). Note the seafloor, $\Theta$ and $\gamma_n$ are from the gridded data product available from Katsumata et al. (2022) and therefore may not be an exact representation of the seafloor depth, (b) BML thickness above the seafloor with color representation as in (a).



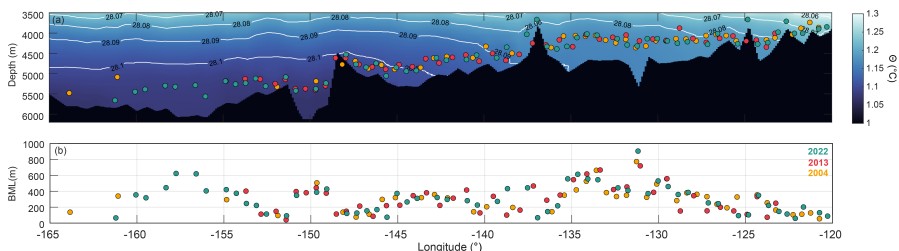

**Figure 6.** The P02 repeat hydrographic line nominally along 30°N with (a) conservative temperature ($\Theta$) with neutral density ($\gamma_n$) as the white contours and the BML thickness for 2002 (green), 2013 (red), and 2004 (orange). Note the seafloor, $\Theta$ and $\gamma_n$ are from the gridded data product available from Katsumata et al. (2022) and therefore may not be an exact representation of the seafloor depth, (b) BML thickness above the seafloor with color representation as in (a).





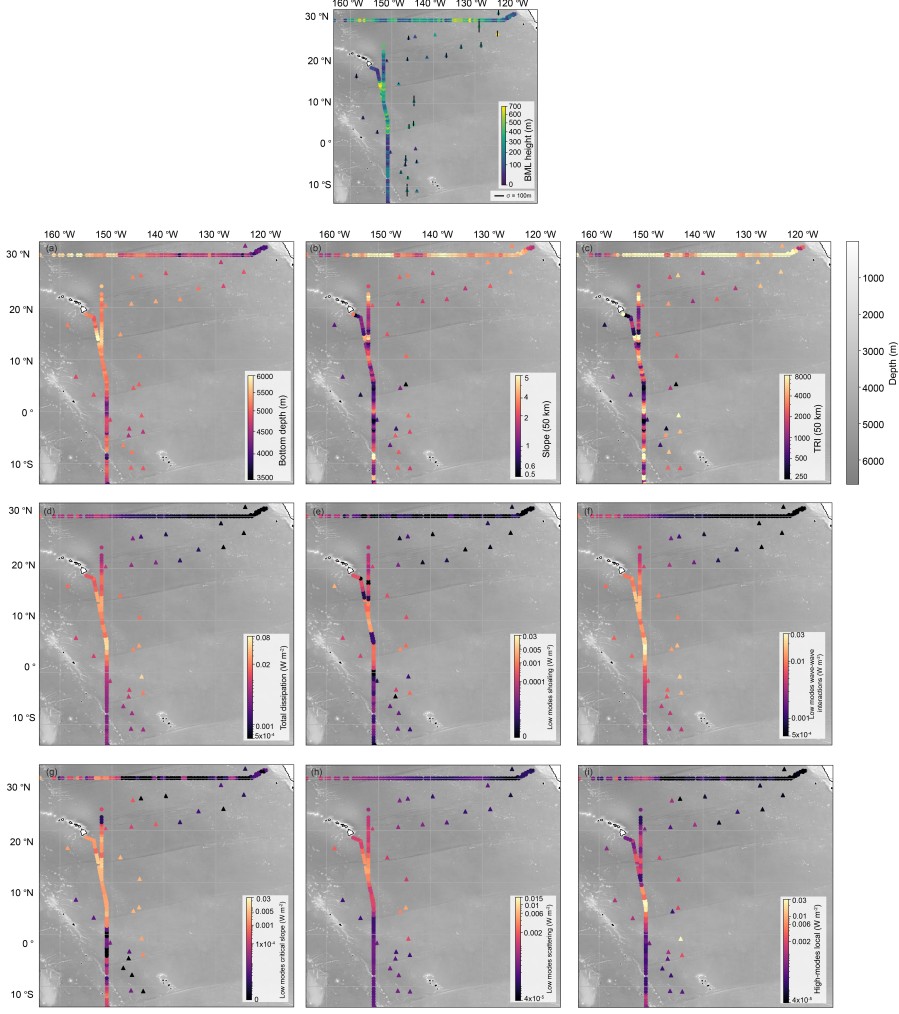

**Figure 7.** All GO-SHIP (circle) and TPT (triangle) site variables used for the Random Forest Regressor (RF). The top BML thickness figure is the same as 3 for reference. (a) Bottom depth, m (b) slope over a 50 km radius, ° (c) terrain roughness index (TRI) over a 50 km buffer (d) total internal tide energy dissipation, W m$^{-2}$ (e) low-mode dissipation from shoaling, W m$^{-2}$ (f) low-mode dissipation from wave-wave interaction, W m$^{-2}$ (g) low-mode dissipation from critical slopes, W m$^{-2}$ (h) low-mode dissipation from scattering, W m$^{-2}$ and (i) high-mode dissipation from local processes, W m$^{-2}$. (d) to (i) are from Lavergne et al. (2019). The background regional bathymetry is from the Global Multi-Resolution Topography (GMRT) Synthesis (Ryan et al. (2009)) Released CC BY 4.0 Deep | Attribution 4.0 International | Creative Commons. Note the colour scale is different in (d) compared with (e)-(i)

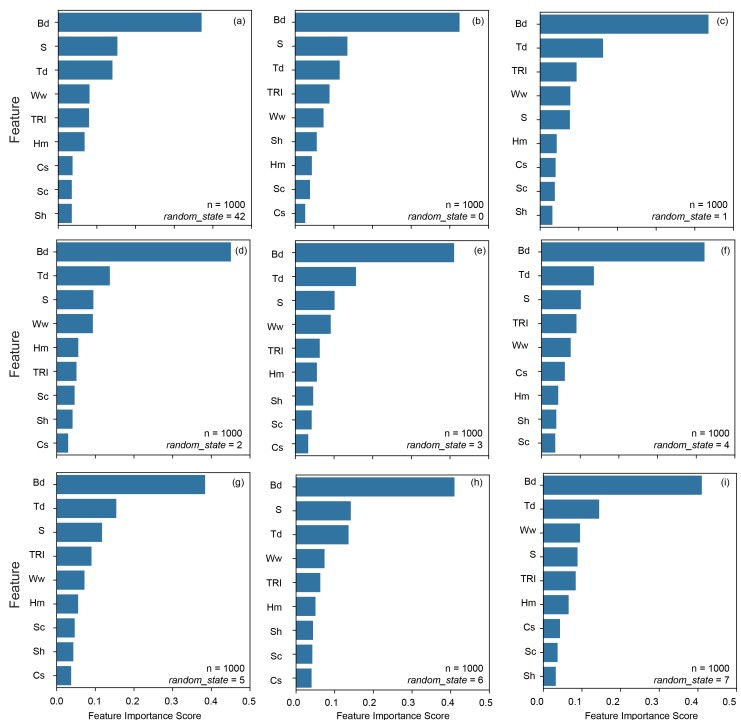

**Figure 8.** Feature importance scores for each feature output from the number of iterations (n) = 1000 for the *train test split random_state* values of the 80-20 data split for (a) 42 and (b) - (i) for 0 to 7 for the Random Forest Regressor. Bd = bottom depth, S = slope, TRI = terrain roughness index, Td = total internal tide energy dissipation, Sh = low-mode dissipation from shoaling, Ww = low-mode dissipation from wave-wave interaction, Cs = low-mode dissipation from critical slopes, Sc = low-mode dissipation from scattering and Hm = high-mode dissipation from local processes.

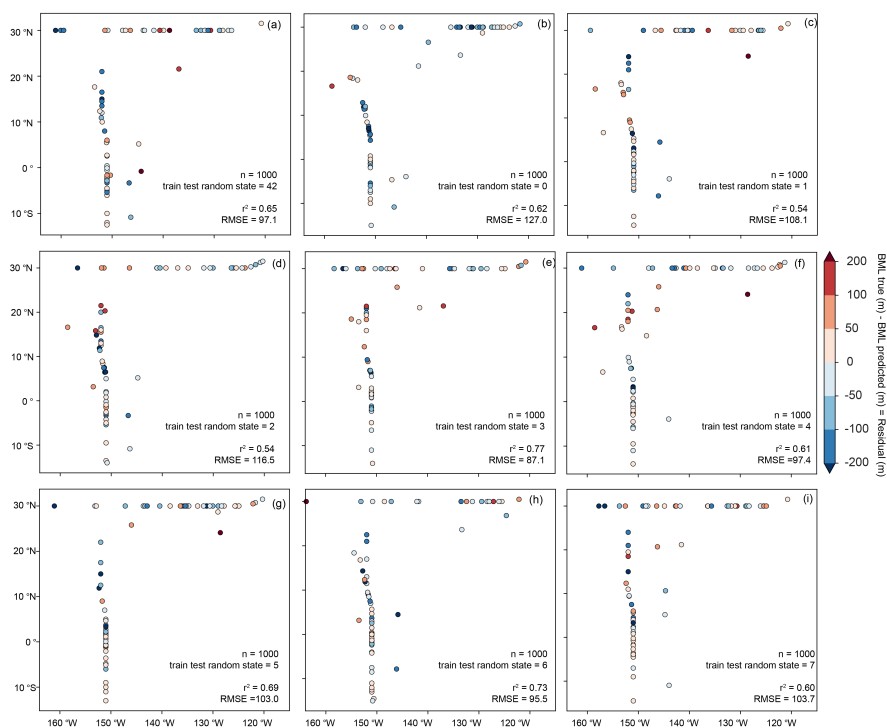

**Figure 9.** Spatial plots of the BML residual (m) (BML true - BML predicted) for the *train test* random_state values used to train the Random Forest Regressor with the number of iterations = 1000, generating a spread of the data for random_state equal to (a) 42 and (b) - (i) for 0 to 7. The correlation coefficient and the Root Mean Squared Error (RMSE) is displayed on each figure.

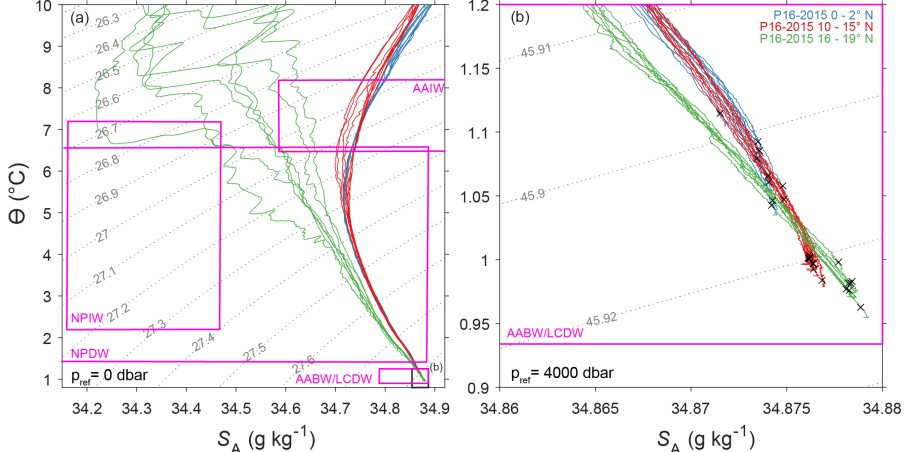

**Figure 10.** Conservative temperature ($\Theta$, °) - Absolute Salinity (SA, g kg⁻¹) plots for latitudinal sections of the P16 line for the 2015 occupation between 0 - 2°N (blue), 10 - 15°N (red) and 16 - 19°N (green). (a) Wider portion of the water column with the (b) limits outlined. In (a) the dashed contour lines show the potential density referenced to the 0 dbar ($\sigma_0$) and in (b) the dashed lines show the potential density referenced to 4000 dbar ($\sigma_4$). $\Theta$-SA North Pacific water mass properties are shown in magenta Fuhr et al. (2021) and the BML for the profile is displayed with a black X. NPIW = North Pacific Intermediate Water, AAIW = Antarctic Intermediate Water, NPDW = North Pacific Deep Water, AABW = Antarctic Bottom Water, used here interchangeably with Lower Circumpolar Deep Water (LCDW). Displayed and calculated with the TEOS-10 toolbox McDougall and Barker (2011)

**Table 1.** Random Forest performance for different numbers of estimators (n), comparing models trained on all features vs. the top five features.

| | All | | | Top 5 | | |
|---|---|---|---|---|---|---|
| **Number of estimators (n)** | **1000** | **500** | **100** | **1000** | **500** | **100** |
| *(b) Train_test random state = 42* | | | | | | |
| $r^2$ | 0.65 | 0.65 | 0.65 | 0.67 | 0.67 | 0.67 |
| RMSE | 97.1 | 97.4 | 95.7 | 93.9 | 93.7 | 93.5 |
| MAE | 71.4 | 72.0 | 71.4 | 70.0 | 69.7 | 69.7 |
| *(b) Train_test random state = 0* | | | | | | |
| $r^2$ | 0.62 | 0.62 | 0.60 | 0.63 | 0.63 | 0.64 |
| RMSE | 127.0 | 127.9 | 129.7 | 125.3 | 125.4 | 124.5 |
| MAE | 86.3 | 87.0 | 86.7 | 85.2 | 85.4 | 85.1 |
| *(c) Train_test random state = 1* | | | | | | |
| $r^2$ | 0.54 | 0.54 | 0.54 | 0.56 | 0.56 | 0.54 |
| RMSE | 108.1 | 108.2 | 109.0 | 106.4 | 106.4 | 108.0 |
| MAE | 72.3 | 73.6 | 73.4 | 71.3 | 70.9 | 72.8 |





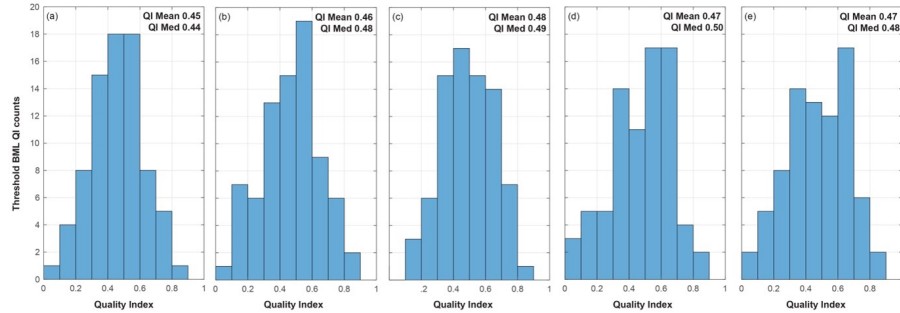

**Figure A1.** Histogram plots of the quality index values from the threshold BML height based on different threshold values of conservative temperature (Θ) as (a) 0.001°C, (b) 0.002°C, (c) 0.003°C, (d) 0.004°C and (e) 0.005°C



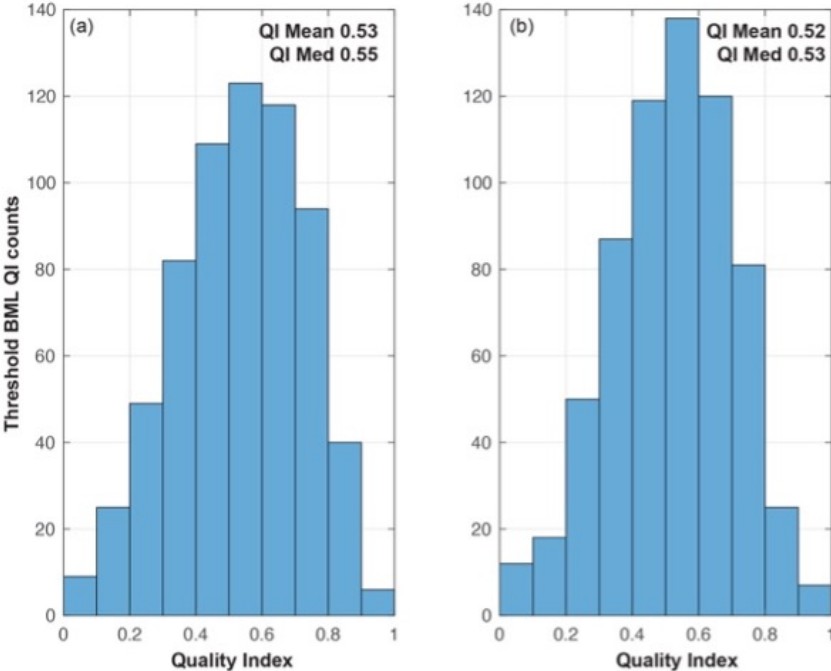

**Figure A2.** Histogram plots of the quality index values from the threshold BML height based on different threshold values of conservative temperature ($\Theta$) as (a) 0.001°C, (b) 0.002°C, (c) 0.003°C, (d) 0.004°C and (e) 0.005°C





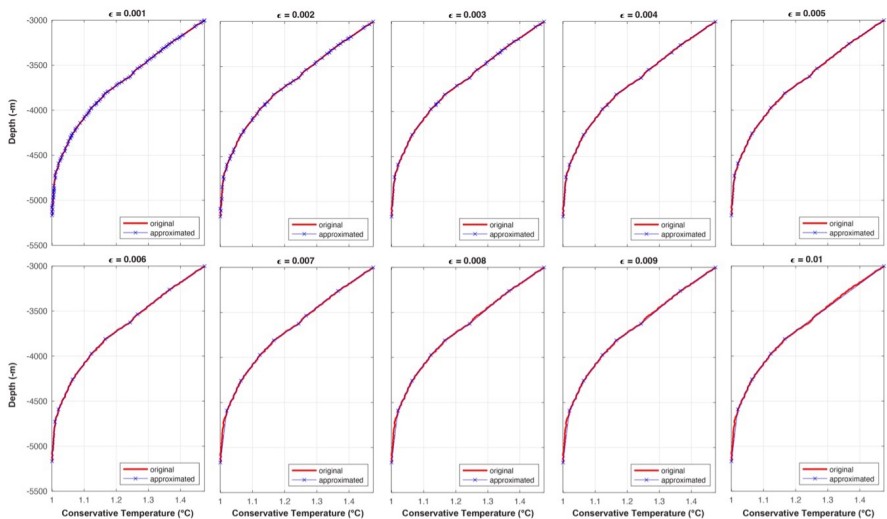

**Figure A3.** Douglas-Peuker Algorithm output (approximated, in blue) for different values of $\varepsilon$ and original profile of $TP2_O M3_5 400$ in red as an example



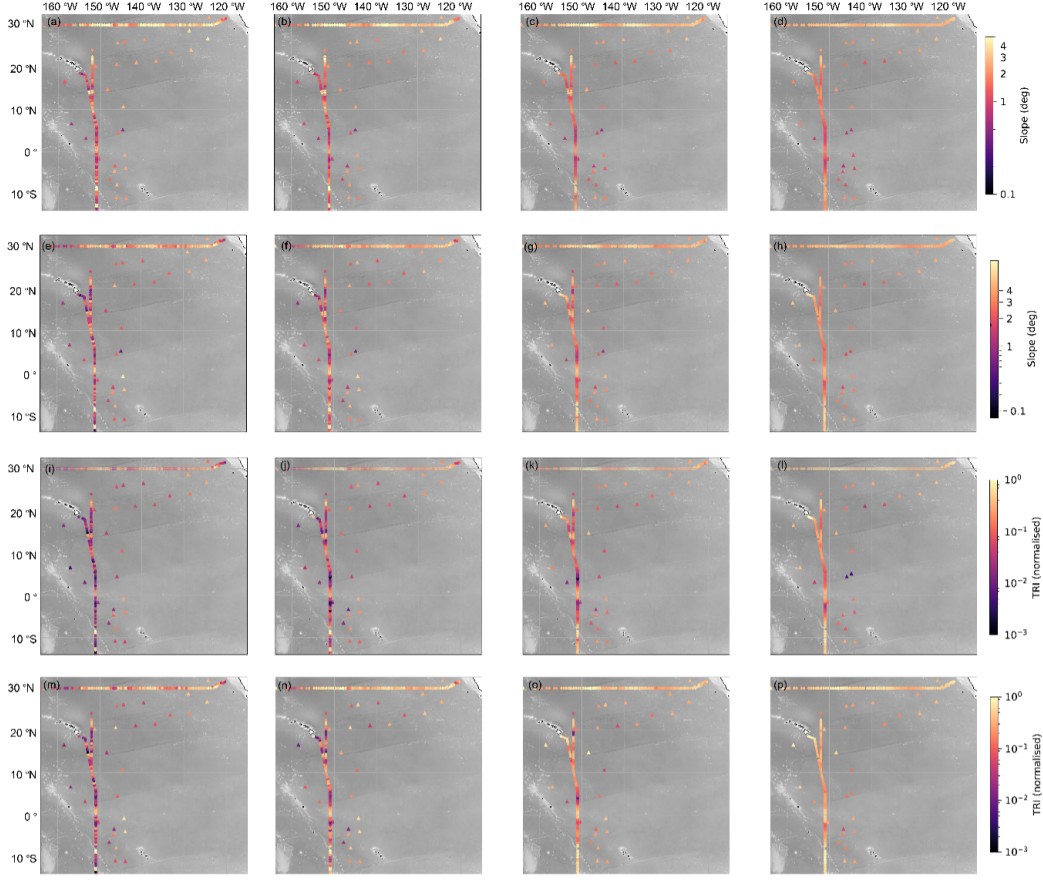

**Figure A4.** GO-SHIP (circle) and TPT (triangle) site variables of the (a) – (d) mean slope over 25, 50, 100 and 200 km buffer zones respectively (deg) and the (e) - (h) standard deviation of slope over the 25, 50, 100 and 200 km buffer zones respectively, (i) – (l) the mean normalised terrain roughness index (TRI) over the 25, 50, 100 and 200 km buffer zones respectively and (m) – (p) displaying the normalised standard deviation of the TRI. For the multiple TPT sites within close proximity ( 3km), TRI and slope are on the centre point. The background regional bathymetry is from the Global Multi-Resolution Topography (GMRT) Synthesis Released CC BY 4.0 Deep | Attribution 4.0 International | Creative Commons Ryan et al. (2009)



Table A1: Station metadata and gaussian mixture model details for each location

| Station | Depth (m) | Lat (°) | Lon (°) | GO-SHIP files | Components | Covariance |
|---|---|---|---|---|---|---|
| TP2_CR1_5200 | 5202 | 17.424 | -151.997 | 73 | 11 | full |
| TP2_CR2_5400 | 5384 | 14.808 | -148.371 | 83 | 14 | full |
| TP2_CR3_5400 | 5310 | 10.635 | -144.672 | 84 | 15 | full |
| TP2_CR4_5000 | 4988 | 5.181 | -144.780 | 62 | 15 | full |
| TP2_CR5_4950 | 4944 | 4.470 | -145.870 | 62 | 15 | full |
| TP2_CR7_4500 | 4588 | -3.926 | -144.014 | 36 | 11 | tied |
| TP2_MA3_5400 | 5140 | 10.630 | -144.689 | 84 | 15 | full |
| TP2_MA4_5000 | 4992 | 5.194 | -144.793 | 62 | 15 | full |
| TP2_MA5_4950 | 4992 | 4.482 | -145.883 | 62 | 16 | tied |
| TP2_MA7_4500 | 4563 | -3.913 | -144.028 | 36 | 13 | tied |
| TP2_OM1_5200 | 5219 | 17.437 | -151.994 | 73 | 13 | full |
| TP2_OM2_5400 | 5385 | 14.791 | -148.376 | 83 | 13 | full |
| TP2_OM3_5400 | 5197 | 10.648 | -144.685 | 84 | 15 | full |
| TP2_OM5_4950 | 4944 | 4.487 | -145.866 | 62 | 16 | tied |
| TP2_OM7_4500 | 4573 | -3.931 | -144.032 | 36 | 11 | tied |
| TP3_CR1_4800 | 4875 | -10.832 | -146.345 | 18 | 11 | tied |
| TP3_CR2_5100 | 5209 | -6.487 | -147.826 | 22 | 18 | tied |
| TP3_CR3_4800 | 4760 | -1.630 | -150.330 | 46 | 13 | tied |
| TP3_CR4_4800 | 4881 | 3.178 | -153.505 | 60 | 9 | full |
| TP3_CR5_4700 | 4816 | 6.649 | -156.946 | 68 | 6 | full |
| TP3_CR7_5200 | 5208 | 16.602 | -158.528 | 75 | 18 | tied |
| TP3_MA1_5100 | 5083 | -10.831 | -146.325 | 18 | 11 | tied |
| TP3_MA2_5200 | 5226 | -6.502 | -147.818 | 22 | 18 | tied |
| TP3_MA3_4800 | 4835 | -1.647 | -150.336 | 46 | 14 | tied |
| TP3_MA4_4800 | 4916 | 3.189 | -153.521 | 60 | 9 | full |
| TP3_MA5_4600 | 4814 | 6.632 | -156.950 | 68 | 6 | full |
| TP3_OM1_5100 | 5101 | -10.817 | -146.337 | 18 | 11 | tied |
| TP3_OM4_4800 | 4873 | 3.169 | -153.521 | 60 | 9 | full |
| TP3_OM5_4600 | 4816 | 6.644 | -156.932 | 68 | 6 | full |
| TP4_CR2_5400 | 5437 | 20.688 | -146.253 | 103 | 17 | tied |

*(Continued on next page)*



*(Continued from previous page)*

| Station | Depth (m) | Lat (°) | Lon (°) | GO-SHIP files | Components | Covariance |
|---|---|---|---|---|---|---|
| TP4_CR3_5400 | 5445 | 21.163 | -141.568 | 24 | 11 | full |
| TP4_CR4_5300 | 5319 | 21.563 | -136.898 | 14 | 10 | full |
| TP4_CR6_4700 | 4792 | 24.123 | -128.540 | 4 | 17 | tied |
| TP4_CR7_4800 | 4873 | 26.825 | -124.635 | 4 | 17 | tied |
| TP4_MA1_5200 | 5229 | 20.315 | -151.210 | 120 | 11 | full |
| TP4_MA2_5400 | 5476 | 20.688 | -146.264 | 103 | 11 | full |
| TP4_MA3_5400 | 5491 | 21.173 | -141.551 | 24 | 11 | full |
| TP4_MA4_5300 | 5335 | 21.560 | -136.917 | 14 | 10 | full |
| TP4_MA5_5000 | 5105 | 23.647 | -133.339 | 5 | 18 | tied |
| TP4_MA6_4700 | 4786 | 24.138 | -128.551 | 4 | 17 | tied |
| TP4_MA7_4800 | 4906 | 26.841 | -124.623 | 4 | 17 | tied |
| TP4_OM1_5200 | 5232 | 20.297 | -151.208 | 120 | 11 | full |
| TP4_OM2_5400 | 5445 | 20.704 | -146.272 | 103 | 17 | tied |
| TP4_OM3_5400 | 5438 | 21.181 | -141.568 | 24 | 11 | full |
| TP4_OM4_5300 | 5367 | 21.577 | -136.909 | 14 | 10 | full |
| TP4_OM5_5000 | 5148 | 23.647 | -133.358 | 5 | 11 | tied |
| TP4_OM6_4700 | 4793 | 24.071 | -128.560 | 4 | 17 | tied |
| TP4_OM7_4800 | 4934 | 26.824 | -124.615 | 4 | 17 | tied |
| TP5_CR1_4300 | 4306 | 31.843 | -124.368 | 4 | 18 | tied |
| TP5_CR2_4600 | 4645 | 28.687 | -129.032 | 4 | 11 | tied |
| TP5_CR5_5200 | 5257 | 25.795 | -145.973 | 77 | 10 | full |
| TP5_MA1_4300 | 4310 | 31.837 | -124.347 | 4 | 18 | tied |
| TP5_MA2_4600 | 4647 | 28.690 | -129.013 | 4 | 11 | tied |
| TP5_MA3_4800 | 4846 | 26.577 | -139.637 | 24 | 11 | full |
| TP5_MA5_5200 | 5262 | 25.810 | -145.982 | 77 | 10 | full |
| TP5_OM1_4300 | 4289 | 31.824 | -124.364 | 4 | 18 | tied |
| TP5_OM2_4600 | 4636 | 28.704 | -129.026 | 4 | 11 | tied |
| TP5_OM5_5200 | 5232 | 25.795 | -145.994 | 77 | 10 | full |
| TP6_CR4_4700 | 4662 | -3.335 | -146.653 | 38 | 13 | tied |
| TP6_CR5_4500 | 4465 | -4.616 | -146.773 | 31 | 12 | full |
| TP6_CR6_5200 | 5140 | -7.822 | -146.141 | 18 | 11 | full |



*(Continued from previous page)*

| Station | Depth (m) | Lat (°) | Lon (°) | GO-SHIP files | Components | Covariance |
|---|---|---|---|---|---|---|
| TP6_CR7_4900 | 4937 | -10.953 | -143.991 | 18 | 11 | tied |
| TP6_MA4_4700 | 4713 | -3.335 | -146.671 | 38 | 13 | tied |
| TP6_MA5_4500 | 4579 | -4.632 | -146.782 | 31 | 12 | full |
| TP6_MA6_5200 | 5172 | -7.822 | -146.159 | 18 | 11 | full |
| TP6_OM4_4700 | 4651 | -3.319 | -146.662 | 38 | 13 | tied |
| TP6_OM5_4500 | 4465 | -4.632 | -146.814 | 31 | 12 | full |
| TP6_OM6_5200 | 5182 | -7.807 | -146.150 | 18 | 11 | full |
| TP6_OM7_4900 | 4921 | -10.969 | -143.982 | 18 | 11 | tied |