# Peer review of "Bottom mixed layer derivation and spatial variability over the central and eastern abyssal Pacific Ocean"

_EGUsphere, 2025_

## Author Comment (AC1)

Reviewer 2 comments – JK (author) replies in red

The authors use a series of historical and novel abyssal observations in the Eastern Pacific and report on the spatial variability of the bottom boundary layer. This is achieved through testing a series of methods to identify the bottom boundary layer. Following this they use a machine learning technique to identify the contributions to that variability from the bottom topography (depth, slope and roughness) and internal tide dissipation. To the best of my knowledge, this manuscript would be the first publication applying this machine learning approach to the bottom boundary layer. I believe that, following some additional work to strengthen the paper as described below, it would be worthy of publication in Ocean Science.
Thank you for your comments.

It is not clear to me why the authors made the choice to infer the salinity using the GMM. If I understood line 112 correctly there is a CTD on the bottom lander system so I expect they have measured salinity? If there is something I missed here, I think it should be made clearer in the text. At the very least, I would like to see some of derived salinity profiles or T-S diagrams with a comparison with what salinity data is available (either from the cruises or from climatology) to give confidence.
The details of the GMM and the model outputs for each site were given in the Appendix. An additional figure showing an example correction will be added to the Appendix to accompany the table of the GMM fits and explanation, aligning with comments from Reviewer 1. The CTD on some landers functioned correctly, however the majority of CTD profiles (>50%) were not correctable for salinity, therefore the RBRduet data of temperature was used and the salinity was modelled as detailed here. We will add a column in Table A1 which details the GMM fits that specifies the difference between the measured salinity from the salinometer (seafloor sample only) and the modelled salinity from the GMM for added confidence. Given the focus on the abyssal layers of the ocean, and therefore lesser seasonal variation in three-dimensional space, we believe this GMM approach is appropriate.

It is useful somewhere to identify that there is potentially an important difference between a mixed layer defined by hydrography and an active mixing layer where turbulence is acting over the layer. This is very important when inferring the importance of the boundary layer in e.g. water mass transformation. The surface mixed layer community have started to address this nuance and we should too.
Yes, this is an important distinction. We will add a sentence at line 76 "While we use a hydrographic definition of the mixed layer, it is important to note that this does not necessarily coincide with the dynamically active mixing layer defined by turbulence; distinguishing these layers is increasingly recognised as essential when inferring mixing intensity and water-mass transformation. We refer to this as the bottom mixed layer..."

The machine learning technique (as I understand it) is simply identifying where the bottom mixed layer height has the same spatial structure as the potential contributions. This leaves me a little concerned about how to interpret the results given the apparent overlap between some of the inputs. For example, the two main components identified (the bottom depth and the total dissipation) have very similar distributions in Figure 7. I think that the uncertainties and limitations of this method need to be discussed more in the text and acknowledged in the interpretation of the results.
Yes, understood and agreed. This has been added at the start of the discussion and a reference (Line 307)
"Because several predictors share spatial structure (e.g., bottom depth and total dissipation), the RF highlights associations rather than uniquely isolating independent physical drivers, and

this limitation should be considered when interpreting the feature importance results (Strobl et al., 2008)"
In addition, this sentence has been added at line 353
"This variability, combined with shared spatial patterns among some predictors, further underscores that the RF approach here is more diagnostic than predictive and should not be interpreted as uniquely isolating mechanistic controls."

Strobl C, Boulesteix AL, Kneib T, Augustin T, Zeileis A. Conditional variable importance for random forests. BMC Bioinformatics. 2008 Jul 11;9:307. doi: 10.1186/1471-2105-9-307. PMID: 18620558; PMCID: PMC2491635.

Additionally, this will be added to line 325 to make it clear that both are not directly linked: Importantly, dissipation patterns in the de Lavergne et al. (2019) dataset are not set by depth, but by the distribution of internal tide energy sources, scattering pathways, and nonlinear wave–wave losses. Deep basins may accumulate low-mode energy and therefore show elevated dissipation, but this reflects remote energy propagation and decay rather than a mechanistic link to depth itself. Therefore, even where depth and dissipation appear spatially aligned in our RF model, this similarity arises from shared basin-scale structure rather than depth acting as a direct driver of turbulent energy loss.

A thought that you might want to bring into your discussion (optional), if the machine learning is identifying the total dissipation as an important control (subject to my concerns above) does that imply that the hydrographic definition of the boundary layer is likely similar to the active mixing layer for the bottom boundary layer for this region. It also implies strong local control of the hydrographic boundary layer thickness rather than control by relatively far-field topography (e.g. set by the sill depth between basins). This could be an important result.
This is an important result and distinction which we have expanded on.
We will add the below to line 349:
"This correspondence between dissipation and hydrographic BML thickness further suggests that, in this region, the threshold-defined BML is not merely a passive hydrographic feature but may effectively capture the vertical extent over which mixing is dynamically active. Such alignment between hydrography and turbulence is rarely shown explicitly for the abyssal ocean and may point to an underappreciated sensitivity of BML structure to the local dissipation field."

Line 129 – 130 > It is not clear to me what this sentence means
See below
Line 133 – 136 > These sentences seem repetitive
We have rephrased this paragraph to encompass these comments (Line 129).
"In three-dimensional space (Θ-p-SA), the abyssopelagic occupies a relatively small volume (Hjelmervik and Hjelmervik (2014)) McDougall and Jackett (2007). Considering this, gaussian mixture models (GMMs) can be applied to automatically group variables of the water column into a distinct number of components, like clusters, revealing consistent patterns in the data (Maze (2017)). GO-SHIP profiles in the study region (detailed in 2.1) were used to predict practical salinity (SP) from 2,500 m to the seafloor. Only profiles collected within the last 5 years and within 10○ latitude and 10○ longitude were used for each site (Pedregosa et al. (2011a); Maze (2017)). Model selection used information-theory criteria, focusing on the covariance type and number of components in the model using the Gaussian Mixture scikit-learn Python package (Pedregosa et al. (2011a)). The maximum number of components was limited to 21. The covariance types were limited to each component having its own general covariance matrix or all components sharing the same general covariance matrix. The elbow

method was used to determine the number of components in the model with a brief examination of the BIC value (Table A1). If the modelled SP output was physically unstable, the next best option was chosen. The modelled seafloor SP was compared with the measured SP from the seafloor water sample analysed on the vessel using an 8400B Autosal Lab Salinometer (Table A1). A further detailed explanation and GMM details for each TPT profile are provided in Appendix A1."

Line 145 > Define what the difference is relative to (seafloor / deepest part of profile / etc)
Added "difference to the seafloor"

Line 195 > Remove remained
Removed

Line 222 > repetition of datasets
Rephrased "The depth for the GO-SHIP sites was taken as the 'bottom depth' variable available in the datasets."

Line 301 > I didn't understand why the importance of dissipation was enhanced here. If I understood correctly both the dissipation and bottom depth are acting in the same direction to reduce the BML so either could be driving the change?
As this is the results, not discussion, this emphasis has been removed, and the sentence is now phrased "This decrease overlaps with a slight decrease in bottom depth (Figure 7a)."
These distinctions and the links between the depth and dissipation are detailed further above in the comments and will be added to the manuscript.

---

## Author Comment (AC2)

Reviewer 2 comments – JK (author) replies in red

In this manuscript (MS), the authors utilize novel field data to reveal the characteristics of bottom mixed layer (BML) thickness in the central and eastern Pacific and identify key controls on BML variability (ocean depth, total internal tide dissipation, slope). The RF analysis in the MS is first application of machine learning to BML thickness, identifies physically intuitive predictors, which is a notable strength. Several issues related to the RF regression and result interpretation need refinement to enhance the manuscript's scientific rigor and impact. The detailed comments are provided below.

Thank you for the positive comments on the novelty of this study.

1. The authors use GMM to predict practical salinity for TPT profiles using nearby GO-SHIP data, but no quantitative comparison is provided between predicted salinity and independent measurements (e.g., discrete water samples from TPT, or collocated GO-SHIP profiles not used for model training).

Thank you for this insightful comment. As this was not the primary purpose of the paper the details were placed in the Appendix, more detail on the GMM is provided there. Alongside the comments made by the other reviewer, an example figure has been added for one of the sites T-S plot showing the profiles. In addition, the average difference and standard deviation between the salinometer value and modelled SP value has been added to Section 2.2.

2. In the MS, the authors tried to characterize and explain the BML patterns in the central and eastern Pacific that covers the fracture zones, but the data used are mainly located in the central Pacific. In the RF regression, the input GO-SHIP data are selected within 10° of the TPT measurements, which could create a data imbalance that risks biasing the model toward GO-SHIP's spatial characteristics.

The datasets used within the RF regression are not within 10° of the TPT measurements. The 10° limit was the data used to correct the TPT salinity measurements are within those bounds for the GMM for predicting practical salinity and multiple occupations and additional GO-SHIP lines of P21 were included within these 10° bounds. This has been added to the Appendix. It is possible that there could be a data imbalance, but the spatial characteristics are within the same region and therefore applicable in this case.

This sentence has been added to line 233 at the end of the methods where the TPT and GO-SHIP sites are being identified "Because TPT salinities were corrected using nearby GO-SHIP profiles, the combined dataset may inherit some spatial imbalance toward the more regularly sampled GO-SHIP sections; however, both datasets occupy the same hydrographic regime and spatial scale, making them appropriate for joint analysis while acknowledging that this imbalance could introduce minor bias in the RF feature relationships."

The extension of the data to the west coast of the US was the reason for including eastern in the title of the manuscript.

3. In the RF regression, stratification and shear (or the state of stability) are important factors for determining the BML thickness, but are not considered as potential influencing parameters.

Thank you for this comment. In original versions of this paper, we included the mean buoyancy frequency within the BML thickness as previously explored by Liu et al. 2023 (Frontiers in Marine Science). However, we chose to exclude this as the goal was to predict the BML thickness with the RF regression spatially using available datasets over the area spatially, therefore having the stratification would imply already having the CTD profiles, from which you could find the BML already.

4. The relationship between BML thickness and depth, total dissipation, slope, etc., may be more intuitively displayed using scatter plots or similar methods.

Scatter plots and basic regression were our first point of call for this analysis and has been completed at other locations. We found it to be unclear to identify the spatial variability, therefore the RF regression was used.

5. Line 119, the authors mentioned Appendix A1, however in the Appendix A, it is about GMM, the information for TPT profiles can not be found.

This was a mistake, it is Table A1.

6. The MS reports key spatial patterns but provides limited physical explanation for these differences. For example, the authors note cooler, saltier AABW near Hawaii vs. fresher NPDW at the equator, but do not explicitly connect this to stratification (a key control on BML mixing). Stronger stratification in AABW regions should suppress mixing and thin the BML—consistent with thinner BML south of Hawaii.

Thank you for this detail. We draw attention to this in lines 368 to 373 where stratification is mentioned several times. We have now made this more explicit in the final line of the paragraph (line 373). "Therefore, the buoyancy gradient remains difficult to overcome, resulting in a thinner BML in AABW regions compared to regions of NPDW at the equator (Weatherly and Martin (1978))."

7. Figure 2: the locations of each profiles are not shown, making it difficult to link BML thickness to regional features. The "visual interpretation" line in Figure 2c is helpful but should be standardized across all subplots for consistency. The x-axis ranges of some subplots are too large to visually identify the BML thickness. Also the solid dots showing BML results of different methods could cover some critical features of the profiles.

The locations are identified in third column of the figure. The annotations of Figure 2(c) were explicitly only included in only one of the figures as with all annotations included, the figure was very cluttered with all annotations included. Where possible to have all BML thickness results still within the x-axis figure range the x-axis will be changed.

8. Equation 1: The variable $h_1$ is not defined. It likely represents the seafloor depth, but this should be explicitly stated.

Yes, correct. This has been added to the sentence below the equation.

9. The first mention of "$\sigma_4$" (Section 2.3) does not explain it is potential density referenced to 4000 dbar.

Line 145 this distinction has been added. "The threshold method (TH) uses the depth at which the difference in either $\Theta$ or potential density referenced to 4000 dbar, $\sigma_4$, is less than a defined threshold value."

10. Line 146: The citation of Hogg et al. (19821) contains a typo.

Changed, this is 1982.

11. Line 149 and 167: Appendix A2 should be Appendix B.

Changed to Appendix B in both instances.

12. Line 159: The authors refer to "Douglas-Peuchker" is a typo.

Changed to Peucker

13. In appendix B, the authors justify the 0.003°C threshold for BML derivation using "highest mean QI" but do not show some example profiles with BML results regarding different thresholds. QI might be high when the mixed layer results are shallower than the actual ones.

Yes, this is possible, however in order to make a more informed decision than most papers on BML thickness reflect, we opted to include the average for all the profiles to remove any visual identification factor and focus only on this method, therefore getting the mean value as highlighted by Figure A2 and A3 and explained in point 14. The histograms for the TPT profiles threshold values were mistakenly excluded and only the two highest performing threshold values for the GO-SHIP datasets were included. This will be rectified in the revision.

14. In Figure A2, there are no subplots (c-e).

Thank you for pointing this out, mistakenly, an appendix plot was not uploaded as explained above. This was for the GO-SHIP profiles only, the results for the TPT profiles have been added.